# Variants of *DNMT3A* cause transcript-specific DNA methylation patterns and affect hematopoiesis

Tanja Božić[1,2,*], Joana Frobel[1,2,*], Annamarija Raic[1], Fabio Ticconi[3], Chao-Chung Kuo[3], Stefanie Heilmann-Heimbach[4], Tamme W Goecke[5], Martin Zenke[1,2] , Edgar Jost[6], Ivan G Costa[3] , Wolfgang Wagner[1,2]

**De novo DNA methyltransferase 3A (DNMT3A) plays pivotal roles in hematopoietic differentiation. In this study, we followed the hypothesis that alternative splicing of *DNMT3A* has characteristic epigenetic and functional sequels. Specific *DNMT3A* transcripts were either down-regulated or overexpressed in human hematopoietic stem and progenitor cells, and this resulted in complementary and transcript-specific DNA methylation and gene expression changes. Functional analysis indicated that, particularly, transcript 2 (coding for DNMT3A2) activates proliferation and induces loss of a primitive immunophenotype, whereas transcript 4 interferes with colony formation of the erythroid lineage. Notably, in acute myeloid leukemia expression of transcript 2 correlates with its in vitro DNA methylation and gene expression signatures and is associated with overall survival, indicating that *DNMT3A* variants also affect malignancies. Our results demonstrate that specific *DNMT3A* variants have a distinct epigenetic and functional impact. Particularly, DNMT3A2 triggers hematopoietic differentiation and the corresponding signatures are reflected in acute myeloid leukemia.**

## Introduction

DNA methylation (DNAm) of CG dinucleotides (CpGs) is a key epigenetic process in cellular differentiation (Broske et al, 2009). Establishment of new DNAm patterns is particularly mediated by the DNA methyltransferases (DNMTs) DNMT3A and DNMT3B (Okano et al, 1999). Both de novo DNMTs are subject to extensive tissue- or developmental stage–specific alternative splicing (Weisenberger et al, 2002) and different variants can be co-expressed in the same cell (Van Emburgh & Robertson, 2011). There is evidence that alternative splicing of DNMTs affects enzymatic activity or binding specificity (Weisenberger et al, 2002; Choi et al, 2011; Duymich et al, 2016), but so far, it is largely unclear if the different variants mediate different DNAm patterns and if they really possess specific functions in development.

DNMT3A is of particular relevance for hematopoietic differentiation. Conditional ablation of exon 19 of *Dnmt3a* in mice was shown to increase the hematopoietic stem cell pool and impairs their differentiation (Challen et al, 2011, 2014). Furthermore, it has been shown that *DNMT3A* is the most frequently mutated gene in clonal hematopoiesis of the elderly and this was linked to a higher risk for hematological malignances—indicating that aberrations in *DNMT3A* play a central role for clonal hematopoiesis (Genovese et al, 2014; Jaiswal et al, 2014; Xie et al, 2014). However, the functional roles of specific *DNMT3A* variants in hematopoietic differentiation and malignancies have not been systematically compared.

Acute myeloid leukemia (AML) is frequently associated with genomic mutations in *DNMT3A*—either at the highly recurrent position R882 (Yamashita et al, 2010) or at other sites within this gene (Ley et al, 2010; Yoshizato et al, 2015). These mutations are associated with poor prognosis and are used for risk stratification in AML (Ley et al, 2010; Ribeiro et al, 2012). In our previous study, we demonstrated that ~40% of AML patients have an aberrant hypermethylation within the *DNMT3A* gene (Jost et al, 2014). This hypermethylation is also associated with poor prognosis in AML (Jost et al, 2014) and myelodysplastic syndromes (Mies et al, 2016) and was therefore termed "*DNMT3A* epimutation". Notably, mutations and the "epimutation" of *DNMT3A* resulted in down-regulation of exons associated with transcript 2 (coding for DNMT3A2) (Jost et al, 2014). Furthermore, in vitro expansion of hematopoietic stem and progenitor cells (HSPCs) resulted in down-regulation of *DNMT3A* transcript 2 (Weidner et al, 2013). In this study, we followed the hypothesis that different isoforms of DNMT3A have distinct molecular and functional sequels and thereby affect hematopoietic differentiation and malignancy.

---

[1]Helmholtz-Institute for Biomedical Engineering, Stem Cell Biology and Cellular Engineering, RWTH Aachen University Medical School, Aachen, Germany    [2]Institute for Biomedical Engineering—Cell Biology, RWTH Aachen University Medical School, Aachen, Germany    [3]Institute for Computational Genomics, RWTH Aachen University Medical School, Aachen, Germany    [4]Institute of Human Genetics, Department of Genomics, Life & Brain Center, University of Bonn, Bonn, Germany    [5]Department of Obstetrics and Gynecology, RWTH Aachen University Medical School, Aachen, Germany    [6]Clinic for Hematology, Oncology, Hemostaseology and Stem Cell Transplantation, RWTH Aachen University Medical School, Aachen, Germany

Correspondence: wwagner@ukaachen.de
*Tanja Božić and Joana Frobel contributed equally to this work as first authors

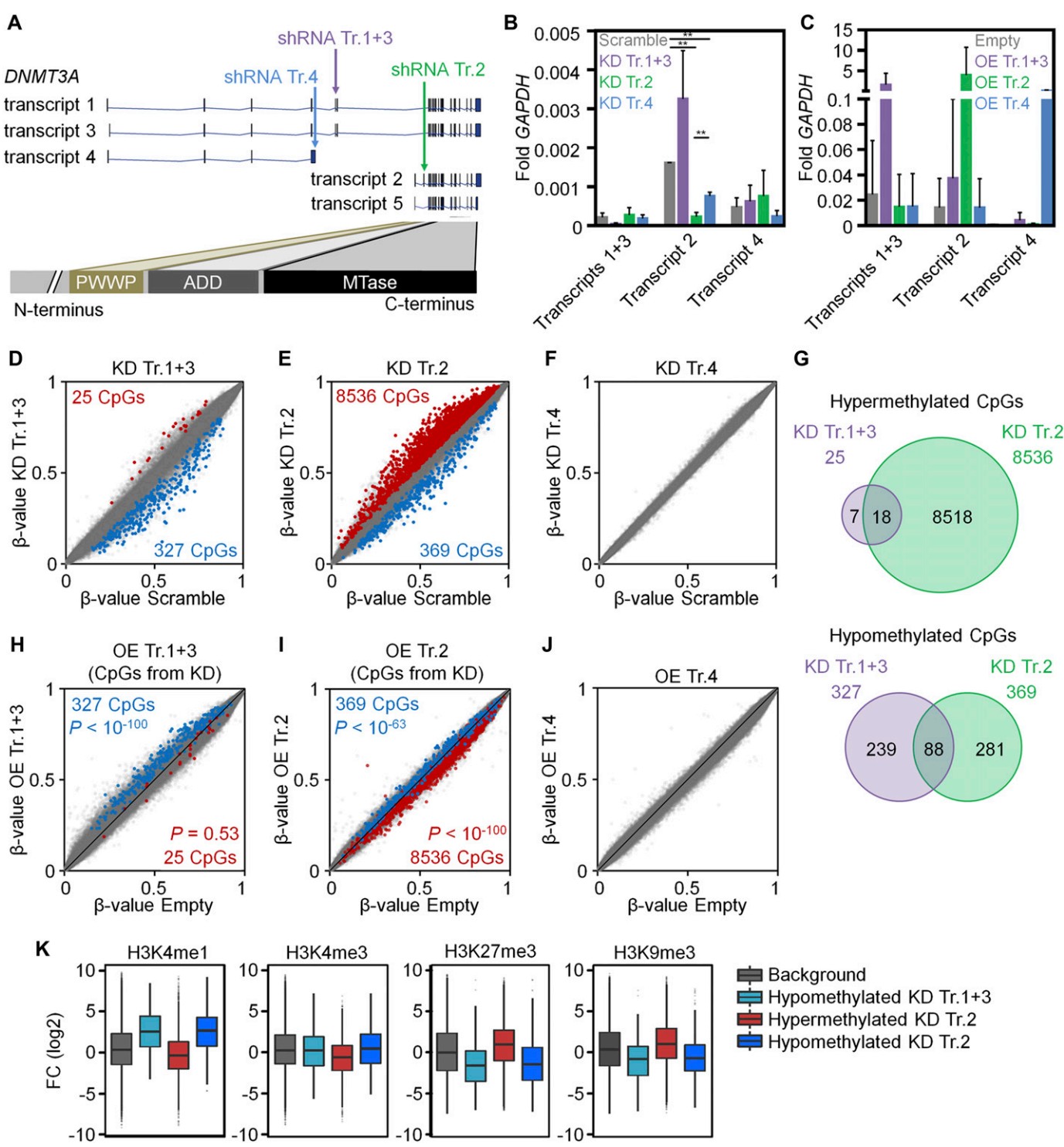

**Figure 1. DNMT3A variants cause unique DNAm signatures.**
**(A)** Schematic representation of five protein-coding *DNMT3A* splice variants and their functional protein domains. The target sites of shRNA are indicated. **(B, C)** KD (B) and OE (C) of individual transcripts was confirmed with RT-qPCR (relative expression versus *GAPDH*; mean ± SD; *n* = 3). **(D–F)** Scatter plots of DNAm profiles upon down-regulation of transcripts 1+3 (D), transcript 2 (E), and transcript 4 (F) compared with scrambled shRNA controls (mean β-values; *n* = 3). Significantly hyper- and hypomethylated CpGs are depicted in red and blue, respectively (adj. *P* < 0.05). **(G)** Venn diagrams demonstrate overlap of DNAm changes upon modulation of transcripts 1+3 and transcript 2. **(H–J)** Scatter plots of DNAm profiles upon OE of transcripts 1+3 (H), transcript 2 (I), and transcript 4 (J), as compared with empty control vectors without any *DNMT3A* transcript (additional independent biological replicates, *n* = 3). Highlights indicate the CpGs that were significantly changed upon KD of the corresponding transcripts (hyper- and hypomethylated upon KD in red and blue, respectively; see Fig 1D and E). Almost all CpGs are modified in opposite directions upon KD and OE and the corresponding *P*-values are indicated (Fisher's *t* test). **(K)** Enrichment of histone marks within 250 bp up- and downstream of each relevant CpG with significant DNAm changes upon KD of transcripts 1+3, transcript 2, and random 50,000 CpGs (Background). Fold change (FC) was calculated over the input background signal. *P < 0.05, **P < 0.01, ***P < 0.001 (*t* test).

# Results and Discussion

### *DNMT3A* splice variants have transcript-specific DNAm signatures

So far, five protein coding transcripts of *DNMT3A* have been described: transcripts 1 and 3 (ENST00000264709.7 and ENST00000321117.9, respectively) have different transcription start sites, but code for the same full-length protein isoform, referred to as DNMT3A1; transcript 2 (ENST00000380746.8) is truncated at the N-terminus, and codes for the protein isoform DNMT3A2; transcript 4 (ENST00000406659.3), codes for DNMT3A4, is truncated at the C-terminus, and lacks the catalytic active methyltransferase (MTase) domain and the PWWP (Pro-Trp-Trp-Pro) and ADD (ATRX-DNMT3-DNMT3L) domains that can interact with various binding partners and chromatin modifications (Yang et al, 2015); and recently, an additional transcript 5 was identified (ENST00000402667.1) that encodes for a similar isoform as DNMT3A2, but lacks the second exon of transcript 2. Although we were able to amplify transcript 5, this transcript does not have a unique exon for transcript-specific knockdown and therefore it was not considered for further analysis.

Initially, individual transcripts were knocked down (KD) in cord blood (CB)–derived CD34$^+$ HSPCs with shRNAs targeting exon 5 of transcripts 1 and 3 (shRNA Tr.1+3), exon 2 of transcript 2 (shRNA Tr.2), and exon 4 of transcript 4 (shRNA Tr.4; Fig 1A). As a control we used a shRNA containing a scrambled sequence. Significant KD was validated by real-time quantitative PCR (RT-qPCR) with primers targeting transcript-specific exons (Figs 1B and S1A; *n* = 3). In addition, *DNMT3A* transcripts 1+3, 2, and 4 were cloned into vectors for constitutive overexpression (OE) and delivered by lentiviral infection into three additional replicates of CD34$^+$ HSPCs. Efficient OE was verified by RT-qPCR and Western blot (Figs 1C and S1B, and C; *n* = 3).

To investigate whether modulation of *DNMT3A* splice variants evokes transcript-specific epigenetic changes, we analyzed global DNAm patterns. KD of transcripts 1+3 resulted in 352 CpGs with significant DNAm changes compared with HSPCs infected with the scrambled control (Fig 1D and Table S1A; *n* = 3; adj. *P* < 0.05) and KD of transcript 2 evoked 8,905 significant DNAm changes (Fig 1E and Table S1B; *n* = 3, adj. *P* < 0.05), whereas KD of transcript 4, which does not comprise the MTase domain, did not result in any significant changes (Fig 1F). Down-regulation of transcripts 1+3 resulted preferentially in hypomethylation, whereas down-regulation of transcript 2 was rather associated with hypermethylation. The latter is counterintuitive, but in line with preferential hypermethylation in reduced representation bisulfite sequencing data of *Dnmt3a*-null HSCs (Challen et al, 2011). It is conceivable that other DNMT3A isoforms or even DNMT3B compensate the down-regulation of *DNMT3A* transcript 2 (Challen et al, 2014). On RNA-level, KD of transcript 2 resulted only in a very moderate up-regulation of transcript 1+3 and transcript 4 (Fig S1A), but the general impact on the stoichiometry of different MTases or functional mechanisms remains to be elucidated. Overall, cross-comparison of differential DNAm upon KD of transcripts 1+3 and KD of transcript 2 revealed a relatively low overlap, indicating that most of the changes were transcript-specific (Figs 1G and S2). This is in line with previous reports indicating that DNMT3A1 is associated with heterochromatin, whereas DNMT3A2 is preferentially associated with euchromatin (Chen et al,

2002), and that the two isoforms have different binding preferences in mouse embryonic stem cells (Manzo et al, 2017). Down-regulation of transcript 2 led to a significant hypermethylation of two CpGs within *DNMT3A* (cg20948740 and cg11354105; Δ*β*-value = 0.0505 and 0.0502, respectively; adj. *P* < 0.05) that are localized close to the epimutation of *DNMT3A*, which is frequently deregulated in AML (Jost et al, 2014). Thus, the *DNMT3A* epimutation may not only result in the down-regulation of transcript 2 (Jost et al, 2014), but also the other way around indicating that DNAm at this region and splicing may be mutually regulated. Furthermore, there is recent evidence that DNMT3A itself interacts with splicing factors and affects global alternative splicing patterns in HSCs (Ramabadran et al, 2017). Such feedback mechanisms can ultimately determine the relative abundance of specific *DNMT3A* transcripts.

Unexpectedly, OE of individual transcripts did not entail significant DNAm changes in comparison with controls (*n* = 3, adj. *P* < 0.05). This might be attributed to unaltered endogenous expression of other *DNMT3A* variants. However, when we analyzed those CpGs with significant DNAm changes upon KD, we observed that the vast majority of hypomethylated CpGs were now hypermethylated upon OE and vice versa (Fig 1H–J). Statistical analysis (Fisher's *t* test) unequivocally demonstrated that down-regulation and OE of specific *DNMT3A* variants have significant opposing and transcript-specific effects on DNAm patterns (transcripts 1+3: hypomethylated CpGs *P* < 10$^{-100}$; and transcript 2: hypermethylated CpGs *P* < 10$^{-100}$, hypomethylated CpGs *P* < 10$^{-63}$), indicating that the effects of the shRNAs are target specific.

To determine whether targets of *DNMT3A* variants are related to the histone code we used chromatin immunoprecipitation sequencing data (ChIP-seq) of short-term cultured human CD34$^+$ cells from the International Human Epigenome Consortium (IHEC). Hypomethylation upon either KD of transcripts 1+3 (327 CpGs) or transcript 2 (369 CpGs) was enriched in genomic regions with the histone mark H3K4me1, typically associated with enhancers. In fact, it has recently been shown that hypomethylation in AML is enriched in active enhancer regions marked with H3K4me1 and H3K27ac (Yang et al, 2016; Glass et al, 2017). In contrast, the 8,536 hypermethylated CpGs upon KD of transcript 2 were associated with repressive histone marks H3K27me3 and H3K9me3 (Figs 1K and S3 and Table S2). This is in line with the finding that, particularly, hypermethylation upon KD of transcript 2 hardly occurred at CpG islands (CGIs) and promoter regions (Fig S4A). Similarly, it has been shown that CpGs associated with shore regions, and not CGIs, play a central role for epigenetic classification of AML (Glass et al, 2017). Motive enrichment analysis of differentially methylated regions (DMRs) of DNMT3A1 (flanking 50 bp around each differentially methylated CpG site) revealed a significant enrichment of binding sites for Spi-B transcription factor (TF) (SPIB; adj. *P* = 5.6 × 10$^{-4}$, Fig S4B), whereas DMRs of DNMT3A2 were particularly enriched in the binding sites for the hematopoietic TFs GATA binding protein 4 (GATA4; adj. *P* = 4.0 × 10$^{-9}$), GATA1 (adj. *P* = 1.5 × 10$^{-8}$), runt related TF1 (RUNX1; adj. *P* = 3.6 × 10$^{-8}$), GATA2 (adj. *P* = 3.6 × 10$^{-8}$), GATA5 (adj. *P* = 3.4 × 10$^{-8}$), GATA3 (adj. *P* = 1.8 × 10$^{-6}$), and SRY-box 10 (SOX10; adj. *P* = 2.0 × 10$^{-6}$; Fig S4C). Gene ontology (GO) classification indicated that hypomethylation upon KD of transcripts 1+3 or transcript 2 occurs preferentially in promoter regions of genes associated with lymphocyte activation (particularly of T cells) or immune regulation, respectively (Fig S4D). Recent findings also

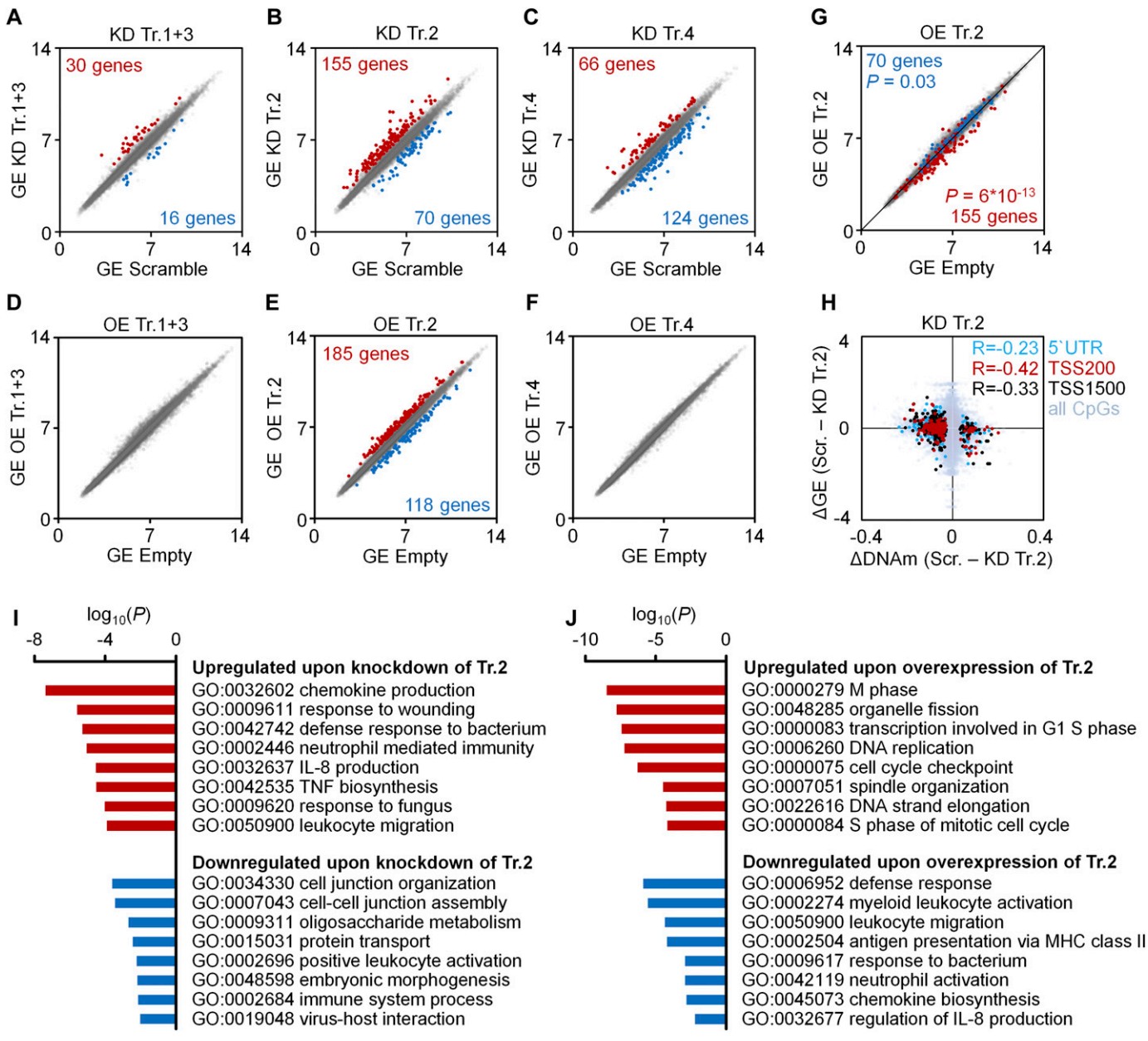

**Figure 2. Gene expression changes upon modulation of *DNMT3A* transcripts.**
**(A–F)** Scatter plots of gene expression (GE; Affymetrix Gene ST 1.0 microarray) upon KD (A–C) or OE (D–F) of transcripts 1+3 (A, D), transcript 2 (B, E), and transcript 4 (C, F), as compared with scrambled shRNA controls or empty control vectors (mean log2 values of normalized data; *n* = 3). Numbers of genes that reached statistical significance (adj. *P* < 0.05) and at least 1.5-fold up- or down-regulation are indicated in red and blue, respectively. **(G)** To visualize that gene expression changes upon KD and OE of transcript 2 occurred preferentially in opposite directions, the scatter plot depicts significant down-regulation (blue) and up-regulation (red) upon KD of transcript 2 in the data for OE of transcript 2 (Fischer's *t* test). **(H)** DNAm changes of CpGs upon KD of transcript 2 were matched to differential expression of the corresponding genes. Hypermethylated CpGs in promoter regions were associated with down-regulation of gene expression and vice versa (R = −0.23, R = −0.42 and R = −0.33, respectively; *n* = 3). **(I, J)** GO analysis was performed to classify gene expression changes upon KD (I) and OE (J) of transcript 2.

indicate that *Dnmt3a* is relevant for normal thymocyte maturation (Kramer et al, 2017).

### Transcript-specific DNAm changes are reflected in differential gene expression

Next, we analyzed whether modulation of *DNMT3A* transcripts resulted in corresponding gene expression changes of HSPCs.

Differentially expressed genes were selected by adjusted *P*-values (adj. *P* < 0.05) and with an additional cutoff of >1.5-fold differential expression (*n* = 3). Significant changes were observed in 46 genes upon KD of transcripts 1+3 (Fig 2A and Table S3A), 225 genes upon KD of transcript 2 (Fig 2B and Table S3B), and 190 genes upon KD of transcript 4 (Fig 2C and Table S3C) in comparison with transfection with scrambled shRNA. OE of transcripts 1+3 and transcript 4 did not result in significant differential gene expression (Fig 2D and F,

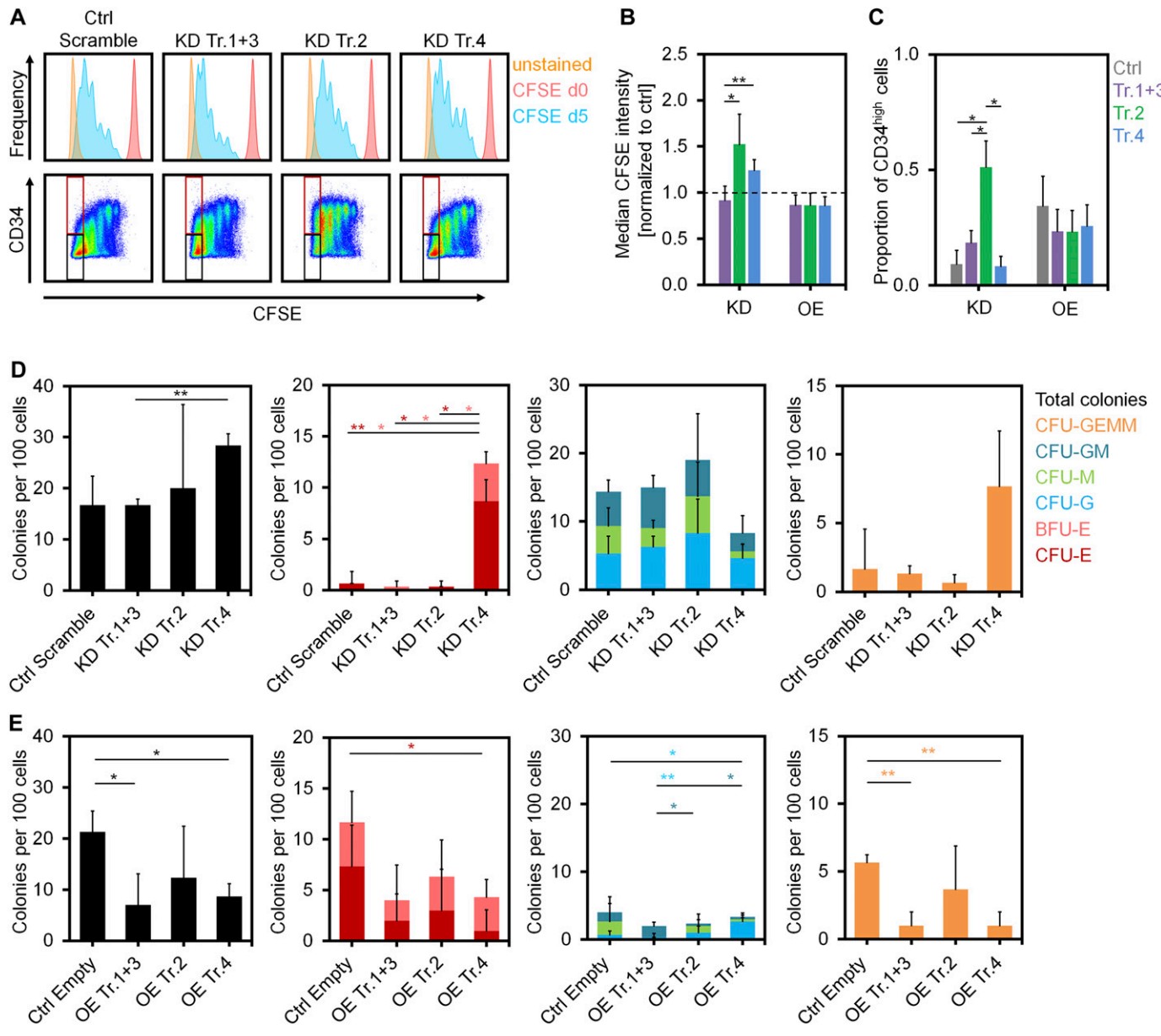

**Figure 3. Impact of *DNMT3A* variants on proliferation and differentiation of HSPCs.**
**(A)** Histograms of residual CFSE staining to estimate proliferation of CD34⁺ cells after transfection with shRNAs after 5 d (blue). For comparison, measurements at day of HSPC isolation (day 0, no cell division, shown in red) and unstained controls at day 5 (shown in orange) are provided. **(B)** KD of transcripts 2 and 4 resulted in higher CFSE retention (slower proliferation) than the control (dashed line; mean ± SD; $n = 3$). **(C)** The proportion of CD34 high cells in the fast proliferating fraction was increased upon KD of transcript 2 (gates are indicated in Fig 3A; mean ± SD; $n = 3$). **(D, E)** CFU frequency (per 100 initially seeded HSPCs) was analyzed after KD (D) and OE (E) of *DNMT3A* transcripts (mean ± SD; $n = 3$). *$P < 0.05$, **$P < 0.01$ ($t$ test).

respectively), which is in line with the moderate DNAm changes. In contrast, OE of transcript 2 resulted in significant gene expression changes of 303 genes (adj. $P < 0.05$; Fig 2E and Table S3D). Notably, when we analyzed the 225 significant genes upon KD of transcript 2 (155 up-regulated and 70 down-regulated), we observed that the vast majority was regulated in the opposite direction upon OE (Fig 2G; Fischer's $t$ test $P = 6 \times 10^{-13}$ and $P = 0.03$, respectively), whereas this was less pronounced for other transcripts.

To determine whether DNAm changes upon KD of transcript 2 are reflected in the corresponding gene expression changes we focused on genes with significant differentially methylated CpGs in the 5′ UTR and up to 200 or 1,500 base pairs upstream of the transcription start sites (TSS200 and TSS1500, respectively). As expected, hypermethylation of promoter regions was associated with down-regulation of gene expression (Fig 2H). Furthermore, GO analysis revealed that differentially expressed genes upon either KD or OE of transcript 2 were enriched in complementary categories for up- and down-regulated genes (Fig 2I and J). Genes down-regulated by DNMT3A2 were enriched in chemokine production (e.g., interleukin 8), immunity, and leukocyte migration; whereas

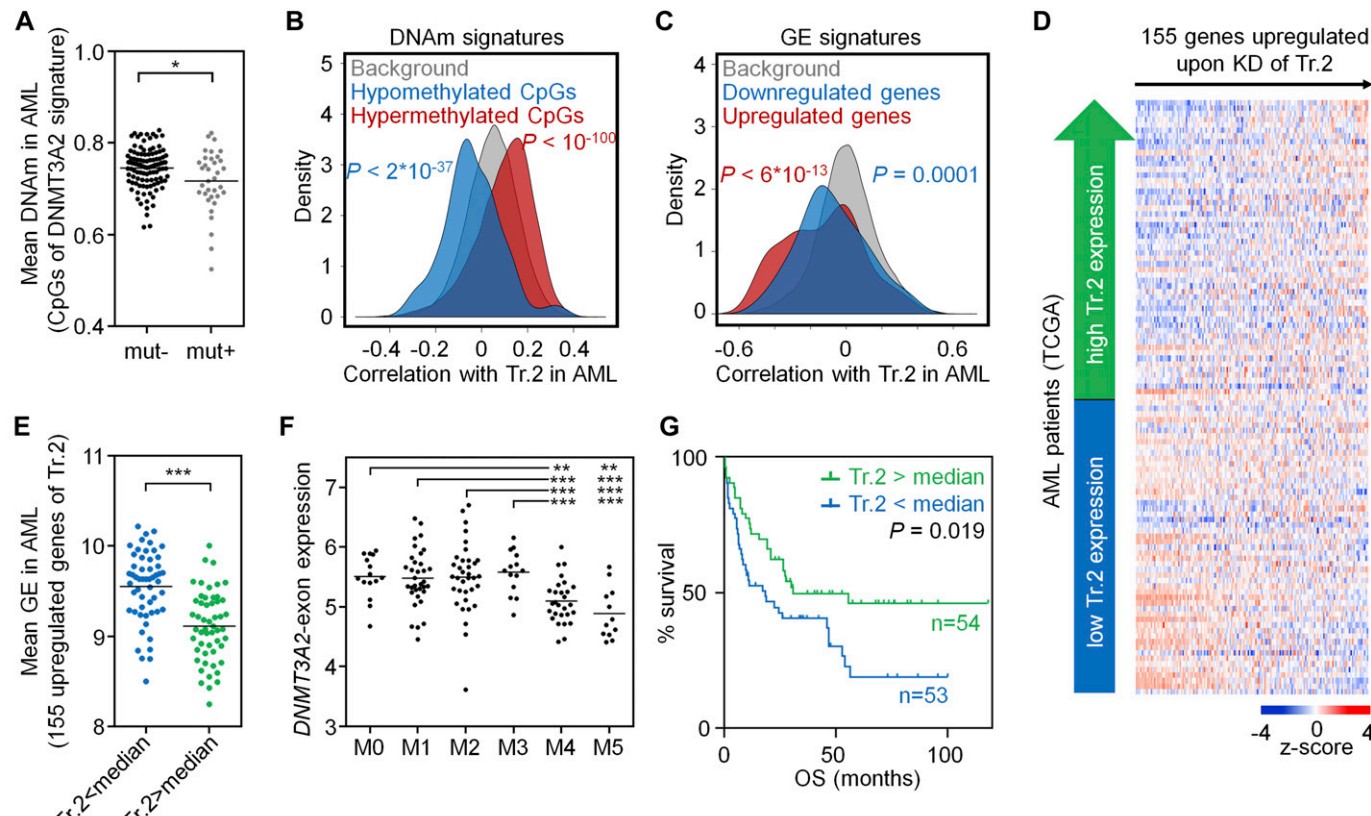

**Figure 4. DNMT3A2 signatures are coherently modified in AML.**
DNMT3A2-associated DNAm and gene expression signatures are recapitulated in the AML dataset of TCGA (The Cancer Genome Atlas Research Network, 2013). **(A)** CpGs that revealed significant DNAm changes upon KD of transcript 2 in HSPCs in vitro (8,905 CpGs, see Fig 1E) had overall significantly lower mean DNAm levels in AML samples with a *DNMT3A* mutation—therefore, samples with *DNMT3A* mutation were excluded from further analysis. **(B)** As a surrogate for transcript 2 expression we used the transcript-specific *DNMT3A2*-exon (ENSE00001486123). CpGs that were either hypo- or hypermethylated upon KD of transcript 2 in HSPCs in vitro, revealed overall highly significant correlation with expression of the *DNMT3A2*-exon in AML. **(C)** In analogy, genes that were differentially expressed upon KD of transcript 2 in HSPCs in vitro were significantly related to expression of the *DNMT3A2*-exon in AML patients. **(D)** Heat map for association of *DNMT3A2* expression in AML with expression of 155 genes that were up-regulated upon KD of transcript 2 in HSPCs. **(E)** These 155 genes were on average significantly higher expressed in AML patients with lower *DNMT3A2*-exon expression (stratified by median). **(F)** The *DNMT3A2*-exon is significantly lower expressed in the AML subgroups M4 and M5 (FAB-classification). **(G)** Kaplan–Meier plot indicates that lower *DNMT3A2*-exon expression (stratified by median) is associated with shorter OS. *P < 0.05, **P < 0.01, ***P < 0.001 (Mann–Whitney test).

up-regulated genes were involved in proliferation. These results indicate that DNMT3A2-associated DNAm changes are overall reflected by the corresponding gene expression changes that are relevant for hematopoietic differentiation.

### *DNMT3A* transcripts impact on hematopoietic differentiation

Different *DNMT3A* variants might be relevant for proliferation and differentiation of HSPCs. CD34$^+$ cells were stained with CFSE after isolation to estimate their proliferation rate based on residual CFSE after 5 d postinfection ($n$ = 3). Proliferation was reduced upon down-regulation of transcript 2 (Fig 3A and B), which is in line with GO enrichment of up-regulated genes upon transcript 2 OE. In contrast, OE of each of the *DNMT3A* transcripts resulted in a moderate increase in proliferation (Fig 3B).

Subsequently, we analyzed whether modulation of *DNMT3A* variants affects maintenance of the surface marker CD34, as a surrogate marker for HSPCs. Down-regulation of transcript 2 maintained CD34 expression even in the fraction of fast proliferating cells with more than five cell divisions ($P$ < 0.05; $n$ = 3;

Fig 3A and C). On the other hand, OE of the *DNMT3A* transcripts reduced the proportion of CD34$^+$ cells in the fast proliferating fraction (Fig 3C). In fact, differential CD34 expression was also observed in additional independent replicates, as well as the corresponding DNAm and gene expression changes, indicating that DNMT3A2 is particularly relevant for loss of CD34 expression during culture expansion of HSPCs (Fig S5).

To investigate whether *DNMT3A* variants affect CFU potential, we seeded HSPCs in methylcellulose at day 5 after infection with shRNAs or OE and analyzed colonies after two additional weeks of culture. Surprisingly, colonies of the erythroid lineage were significantly increased upon KD of transcript 4 (Fig 3D) and reduced upon OE of transcript 4 (Fig 3E). Thus, transcript 4 seems to affect lineage-specific hematopoietic differentiation, although it does not comprise the functional MTase domain. It is conceivable that effects of this isoform are mediated by other epigenetic or transcriptional modifiers by binding to the DNA with the N-terminal region (Suetake et al, 2011). Posttranslational modification of the *Dnmt3a* N-terminus was also shown to facilitate additional interactions (Chang et al, 2011). Overall, our data suggest that specific

*DNMT3A* variants have different effects on proliferation and differentiation of HSPCs in vitro.

### Signatures of DNMT3A2 are recapitulated in AML

To determine whether variant-specific signatures of DNMT3A are also coherently modified in vivo we used the AML dataset of The Cancer Genome Atlas (TCGA) (The Cancer Genome Atlas Research Network, 2013). It has been demonstrated that AML cells with the R882H mutation have severely reduced de novo MTase activity and focal hypomethylation at specific CpGs (Russler-Germain et al, 2014). Therefore, we reasoned that the expression of *DNMT3A* transcripts in AML might also be reflected in our transcript-specific DNAm and gene expression signatures. As a surrogate for expression of individual transcripts, we analyzed expression of transcript-specific exons that were targeted by shRNAs in the KD experiments.

Initially, we analyzed whether *DNMT3A* mutations in AML affect the DNAm signature of DNMT3A2. To this end, we specifically focused on the CpGs that were differentially methylated upon KD of transcript 2 in HSPCs in vitro. AML samples with *DNMT3A* mutations revealed a significantly lower DNAm level in these DNMT3A2-associated CpGs than in those without *DNMT3A* mutations ($P$ = 0.03; Fig 4A) and these samples were therefore excluded for further analysis. Notably, *DNMT3A2*-exon expression in AML patients revealed a highly significant correlation with DNAm levels at CpGs that were hyper- and hypomethylated upon KD of transcript 2 in vitro ($P < 10^{-100}$ and $P = 2 \times 10^{-37}$, respectively; Fig 4B). In analogy, *DNMT3A2*-exon expression in AML was also significantly correlated to expression of the 155 up-regulated and the 70 down-regulated genes of the in vitro transcript 2 signature ($P = 6 \times 10^{-13}$ and $P$ = 0.0001, respectively; Fig 4C and D). Furthermore, the average expression level of these 155 genes was significantly higher in those AML patients with below-median expression of the *DNMT3A2*-exon ($P < 0.0001$; Fig 4E). Similar results were observed when we used a relative expression of the *DNMT3A2*-exon normalized by the overall *DNMT3A* expression level. In analogy, we analyzed the signatures of transcripts 1+3 and transcript 4, but they did not correlate in AML, which might also be because of the relatively small number of differentially expressed genes (Fig S6). Taken together, expression of *DNMT3A2*-exon is associated with variant-specific molecular signatures in AML.

Next, we analyzed whether the expression of *DNMT3A* or its splice variants might also be of clinical relevance in AML (Fig S7). In fact, expression of the *DNMT3A2*-exon was significantly lower in the AML subgroups M4 and M5 of the French–American–British (FAB) classification (Fig 4F). Furthermore, it was lower in patients with poor and intermediate cytogenetic risk score as compared with patients with favorable risk score ($P < 0.01$ and $P < 0.001$, respectively; Fig S7D). Kaplan–Meier analysis ($P = 0.019$; Fig 4G) and Cox regression analysis ($P = 0.016$) indicated that AML patients with a lower expression of the *DNMT3A2*-exon have a significantly shorter overall survival (OS). In tendency, a similar effect was observed for expression of the *DNMT3A1*-exon (Fig S6G), whereas expression of the *DNMT3A4*-exon did not correlate with survival. In comparison to the established molecular parameters for AML stratification, expression of *DNMT3A* variants has a lower prognostic value, but our results support the notion that alternative splicing of *DNMT3A* is also relevant for the disease.

Targeting of DNMTs to specific sites in the genome is orchestrated by a complex interplay with other proteins, TFs, the histone code, and long non-coding RNAs (Yang et al, 2015; Kalwa et al, 2016). The results of this study add a new dimension to this complexity. The 27 exons of *DNMT3A* can be spliced into a multitude of different transcripts—although so far only five protein coding transcripts have been described. Our results demonstrate that different *DNMT3A* variants indeed have different transcript-specific molecular sequels that affect hematopoietic differentiation and malignancy.

## Materials and Methods

### Cell culture of HSPCs

Umbilical CB was obtained after written consent according to guidelines approved by the Ethics Committee of RWTH Aachen Medical School (EK 187-08). CD34⁺ HSPCs were isolated from fresh CB using the CD34 Micro Bead Kit (Miltenyi Biotec) and cultured in StemSpan serum-free expansion medium (Stemcell Technologies) supplemented with 10 μg/ml heparin (Ratiopharm), 20 ng/ml thrombopoietin (PeproTech), 10 ng/ml stem cell factor (PeproTech), 10 ng/ml fibroblast growth factor 1 (PeproTech), and 100 U/ml penicillin/streptomycin (Lonza) (Walenda et al, 2011).

### Lentiviral KD and OE of *DNMT3A* variants

To KD *DNMT3A* transcripts, we designed shRNAs to target transcript-specific exons: exon 5 of transcripts 1+3 (ENSE00001486208), exon 2 of transcript 2 (ENSE00001486123), and exon 4 of transcript 4 (ENSE00001559474; Fig 1A). In brief, forward and reverse oligonucleotides (Metabion; Table S4) were joined and ligated into the pLKO.1 vector (Addgene).

For constitutive OE, the *DNMT3A* transcripts were amplified from cDNA of human blood cells with the High-Capacity cDNA Reverse Transcription Kit (Applied Biosystems) using various combinations of exon-specific primers (Table S5) and cloned into the pLJM1-EGFP (Addgene) by replacing the *EGFP* gene. Successful cloning was validated by sequencing. About 200,000 HSPCs were infected 1 d after isolation and selected by treatment with puromycin (2.5 μg/ml; S-Aldrich) at day 2 after infection.

### Quantitative RT-PCR

KD or OE of *DNMT3A* variants was analyzed by RT-qPCR using the StepOne™ Instrument (Applied Biosystems). RNA was isolated at day 12 after infection, reverse transcribed, and amplified using the Power SYBR Green PCR Master Mix (Applied Biosystems) with transcript-specific primers (Table S6). Gene expression was normalized to *GAPDH*.

### Western blot analysis

OE of *DNMT3A* variants was validated on protein level with Western blot at day 13 after infection of HSPCs. Cells were lysed in RIPA cell lysis buffer, and if available, up to 50 μg of protein lysate was

subjected to SDS–PAGE (10% SDS-polyacrylamide gels). Membranes were blocked for 1 h with 4% skim milk (Sigma-Aldrich) in TBS-T at room temperature and subsequently incubated with primary antibodies for the N-terminal part of DNMT3A (rabbit anti-DNMT3A, 2160, 1:1,000; Cell Signaling Technology) and actin (mouse anti-actin antibody, clone AC-74, 1:10,000; Sigma-Aldrich) at 4°C overnight. Peroxidase-conjugated secondary antibodies (goat anti-rabbit IgG, ab97051, 1:5,000; Abcam; sheep anti-mouse, NA931V, 1:10,000; GE Healthcare) were incubated for 1 h at room temperature and subjected to chemiluminescence using the SuperSignal West Dura Kit (Thermo Fischer Scientific).

### DNAm profiles

DNAm profiles were analyzed for KD and OE condition, each in three independent biological replicates. Genomic DNA was isolated at day 12 after lentiviral infection with the NucleoSpin Tissue Kit (Macherey-Nagel). For DNAm analysis, we have chosen the Infinium HumanMethylation450 BeadChip (Illumina) that covers about 480,000 representative CpG sites at single-base resolution (including 99% of RefSeq genes and 96% of CGIs) (Bibikova et al, 2011). In comparison to genome-wide analysis with reduced bisulfite sequencing data each of these CpGs is detected in all samples with relatively precise estimates of DNAm levels. Furthermore, this microarray platform enabled straightforward comparison with datasets of TCGA. Raw data are deposited at Gene Expression Omnibus (GEO) under accession number GSE103006.

For further analysis of DNAm profiles, we excluded CpGs in single nucleotide polymorphisms and CpGs with missing values in several samples. Significance of DNAm was calculated in R using limma paired $t$ test (adjusted for multiple testing; $P < 0.05$). Association of DMRs with the histone code was analyzed in ChIP-seq data from cultured CD34[+] as provided by the IHEC portal. To estimate differences in levels of histone modifications (active: H3K4me3; enhancer: H3K4me1; repressive: H3K27me3 and H3K9me3), we calculated the read count of a certain histone mark within a 500-bp window around each differentially methylated CpG upon KD of transcripts 1+3 and transcript 2 as compared with 50,000 random CpGs on the microarray. The input read count was calculated from ChIP-seq input signal of a control experiment without antibodies. The read counts were quantile normalized and represented as fold change over the input background signal. Statistics was calculated by the two-tailed $t$ test followed by multiple test correction. Enrichment of CpGs in relation to CGIs or gene regions was based on the Illumina annotation, and significance was estimated by hypergeometric distribution. Enrichment of short, core DNA-binding motifs of TFs within 100 bp around the differentially methylated CpG sites was performed in Python using the Regulatory Genomics Toolbox package (http://www.regulatory-genomics.org/motif-analysis/). GO analysis was performed with the GoMiner software (http://discover.nci.nih.gov/gominer/) of genes associated with differentially methylated CpGs located in the promoter or 5′ UTR regions. Enrichment of specific categories was calculated by the one-sided Fisher's exact $P$-value using all genes represented on the array as a reference.

### RNA expression profiles

Gene expression profiles were analyzed for KD and OE condition, each in three independent biological replicates. Total RNA was isolated at day 12 after infection with the NucleoSpin RNA Kit (Macherey-Nagel) and analyzed with the Affymetrix Human Gene ST 1.0 platform (Affymetrix). Gene expression profiles were deposited at GEO under accession number GSE103007. Raw data were normalized by RMA (Affymetrix Power Tools). Differentially expressed genes were filtered by at least 1.5-fold differential mean expression levels and adjusted $P < 0.05$, which were calculated in R using limma paired $t$ test. GO analysis was performed with the GoMiner software (http://discover.nci.nih.gov/gominer/).

### Flow cytometric analysis

CFSE was used to monitor the number of cell divisions (Walenda et al, 2010). The immunophenotype of HSPCs was examined at day 5 after infection by staining with CD13-PE (clone WM15), CD34-APC (clone 581), CD38-PE (clone HIT2), CD45-V500 (clone HI30), CD56-PE (clone B159; all by Becton Dickinson), CD33-APC (clone AC104.3E3), and CD133/2-PE (clone AC141; both Miltenyi Biotec) with or without CFSE staining. Cells were analyzed using a FACS Canto II (BD) with FACS Diva software (BD).

### CFU assay

HSPCs were infected with lentivirus and selected with puromycin as described above and expanded for 5 d before CFU assay. Subsequently, 100 cells per well were seeded in methylcellulose-based medium (HSC-CFU lite with EPO; Miltenyi Biotec). Granulocyte (CFU-G), macrophage (CFU-M), granulocyte/macrophage (CFU-GM), erythroid (BFU-E and CFU-E), and mixed colonies (CFU-GEMM) were counted according to manufacturer's instructions after 14 d (Walenda et al, 2011).

### Analysis of AML datasets

DNAm and gene expression signatures of *DNMT3A* variants were further analyzed in AML patients of TCGA portal (The Cancer Genome Atlas Research Network, 2013). We focused on 142 patients with data available for HumanMethylation450 Bead Chips, RNA sequencing, and whole exome sequencing. For comparison of DNAm data of the DNMT3A2 signature, only 7,074 CpGs (6,747 hyper- and 327 hypomethylated) of the 8,905 significant differentially methylated CpGs were provided by the TCGA repository. Raw gene expression data were normalized using the variance stabilizing transformation. Expression levels of transcript-specific exons (ENSE00001486208 for transcripts 1+3; ENSE00001486123 for transcript 2, and ENSE00001559474 for transcript 4) were used as a surrogate for the corresponding transcript expression. Alternatively, we normalized the expression of transcript-specific exons to the mRNA level of *DNMT3A* and these relative expression levels provided similar results. Correlations with the signatures were calculated in R. For survival analysis, we used Kaplan–Meier (K-M) and Cox proportional hazards model. For K-M analysis, patients were stratified into two groups according to the median expression

level of the transcript-specific exon. We defined OS as the survival time from the first day of diagnosis to day of death by any cause. K-M plots and Mann–Whitney statistics were generated with GraphPad Prism 6.05 (GraphPad Software) and Cox proportional hazard model was calculated in R.

## Dataset availability

The DNAm and gene expression datasets supporting the conclusions of this article are available in the GEO, under the accession numbers GSE103006 and GSE103007 (part of the Super Series GSE103008).

# Supplementary Information

# Acknowledgments

This work was supported by the Interdisciplinary Center for Clinical Research within the Faculty of Medicine at the RWTH Aachen University (O1-1), by the Else Kröner-Fresenius-Stiftung (2014_A193), by the German Research Foundation (WA 1706/8-1), and by the German Ministry of Education and Research (01KU1402B).

## Author Contributions

T Božić: formal analysis, validation, investigation, visualization, methodology, project administration, and writing—original draft, review, and editing.
J Frobel: formal analysis, validation, investigation, visualization, methodology, project administration, and writing—review and editing.
A Raic: investigation, methodology, and writing—review and editing.
F Ticconi: data curation, formal analysis, and writing—review and editing.
CC Kuo: data curation, formal analysis, and writing—review and editing.
S Heilmann-Heimbach: data curation and writing—review and editing.
TW Goecke: resources and writing—review and editing.
M Zenke: conceptualization, resources, and writing—review and editing.
E Jost: conceptualization, supervision, funding acquisition, and writing—review and editing.
IG Costa: conceptualization, formal analysis, and writing—review and editing.
W Wagner: conceptualization, resources, supervision, funding acquisition, validation, project administration, and writing—original draft, review, and editing.

## Conflict of Interest Statement

RWTH Aachen Medical School has applied for a patent on the DNMT3A epimutation. W Wagner is involved in Cygenia GmbH that can provide service for epigenetic diagnostics. Apart from this, the authors do not have a conflict of interest to declare.

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
