## [Reviewer comments · Life Science Alliance]

Life Science Alliance

Variants of DNMT3A cause transcript-specific DNA methylation patterns and affect hematopoiesis

Tanja Božić, Joana Frobel, Annamarija Raic, Fabio Ticconi, Chao-Chung Kuo, Stefanie Heilmann-Heimbach, Tamme Goecke, Martin Zenke, Edgar Jost, Ivan Cost, and Wolfgang Wagner

DOI: [10.26508/lsa.201800153](https://doi.org/10.26508/lsa.201800153)

Corresponding author(s): Wolfgang Wagner, RWTH Aachen University Medical School

Review Timeline:

Submission Date:	2018-08-10
Editorial Decision:	2018-08-23
Revision Received:	2018-11-20
Editorial Decision:	2018-11-29
Revision Received:	2018-12-04
Accepted:	2018-12-05

Scientific Editor: Andrea Leibfried

Transaction Report:

August 23, 2018

Re: Life Science Alliance manuscript #LSA-2018-00153-T

Prof. Wolfgang Wagner
RWTH Aachen University Medical School
Stem Cell Biology and Cellular Engineering
Helmholtz-Institute for Biomedical Engineering
Pauwelsstrasse 20
Aachen 52074
Germany

Dear Dr. Wagner,

Thank you for submitting your manuscript entitled "Variants of DNMT3A cause transcript-specific DNA methylation patterns and affect hematopoiesis" to Life Science Alliance. The manuscript was assessed by expert reviewers, whose comments are appended to this letter.

As you will see, the reviewers appreciate your data, and they provide constructive input on how to revise it to allow acceptance at Life Science Alliance. Importantly, additional controls are needed (both reviewers) as well as a slight extension of your analysis / a re-analysis of some of the data already at hand (reviewer #1).

These requests seem straightforward to address in a normal revision time, and we would thus be happy to publish such a revised version in Life Science Alliance. We are therefore inviting you to submit a revised version addressing the concerns of the reviewers.

Thank you for this interesting contribution to Life Science Alliance. We are looking forward to receiving your revised manuscript.

Sincerely,

- A letter addressing the reviewers' comments point by point.
- An editable version of the final text (.DOC or .DOCX) is needed for copyediting (no PDFs).
- High-resolution figure, supplementary figure and video files uploaded as individual files: See our detailed guidelines for preparing your production-ready images, <http://life-science-alliance.org/authorguide>
- Summary blurb (enter in submission system): A short text summarizing in a single sentence the study (max. 200 characters including spaces). This text is used in conjunction with the titles of papers, hence should be informative and complementary to the title and running title. It should describe the context and significance of the findings for a general readership; it should be written in the present tense and refer to the work in the third person. Author names should not be mentioned.

B. MANUSCRIPT ORGANIZATION AND FORMATTING:

Full guidelines are available on our Instructions for Authors page, <http://life-science-alliance.org/authorguide>

Reviewer #1 (Comments to the Authors (Required)):

In the study the authors investigate the role of different isoforms of DNMT3a on DNA methylation patterns and gene expression signatures in human stem progenitor cells (CD34+) isolated from the umbilical cord blood. To investigate the function of individual DNMT3a variants the study employs shRNA-mediated knock down (KD) of isoform-specific exons and a lentiviral DNMT3a isoform over-expression (OE) strategy. Overall the study is well designed and addresses the important question if different DNMT3a isoforms show target specific functions, and I am very positive about looking at another version that addresses the following points.

The study would strongly benefit if the authors would show the protein expression levels of the various isoforms in HSPC steady state. From the presented data it is unclear how the different isoforms are expressed in the cells of interest (this is also not visible from the RT-qPCRs). The information on variant abundances should be also included for the KD and OE strategies, rather than showing relative % of expression levels in Fig.1A and B. If western blot analysis are not feasible due to low cell numbers, the expression levels of the individual transcript should be shown relative to a house keeping gene for all experiments. However, Western blots that indicate the levels of all protein isoforms (using general or isoform-specific antibodies) would be very helpful to assess if in the KD/OE experiments the other protein forms would change in their abundance due to changes in isoform stoichiometries.

Interestingly, the study identifies several differentially methylated CpGs upon KD of DNMT3A1 and 3A2. While there is minimal overlap for the detected hyper- and hypo-methylated CpGs in the distinct KD samples (Fig.1G) the authors could also cross-compare the overlap between hypermethylated CpGs after DNMT3a1 KD compared to hypomethylated CpGs in DNMT3a2 KD and vice versa. I am curious to see if the CpGs that reduce methylation in the DNMT3A1 KD overlap with the CpGs that gain in the DNMT3A2 KD - this could potentially hint to a gain of function of the longer isoform.

In figure 1K & L, it is hard to see how significant these differences are, especially for H3K9me3 and H3K27me3 where the effect size seems very low according to the axis labels. I agree that the p values indicate statistical significance, but given the high statistical power used in the test, these are meaningless. I suggest removing the p values altogether and instead showing an additional density for foreground signals for the respective modification (i.e. over all detected peaks). Alternatively a ranked heat map representation, rather than average densities could be shown for the regions around the respective CpGs.

In the end of the publications the authors investigate if variant-specific signatures of DNMT3a can be also found in AML samples. The authors claim that they find only significant correlations with DNMT3A2 isoform, still it is very important to include the results for DNMT3A1 throughout entire Figure 4 in order to make this claim.

Additionally, for the results where the study investigate the clinical relevance of DNMT3A2 abundance, it would be interesting to see if DNMT3A1 shows the same trend and how the overall DNMT3a abundance changes in FAB classifications and in patients with poor and intermediate cytogenetic risk compared to favorable risk scores.

Reviewer #2 (Comments to the Authors (Required)):

1. Short summary of the paper: Bozic et al. present a study investigating the role of distinct isoforms of DNMT3A in hematopoietic stem cell differentiation. Given that DNMT3A mutations are recurrently found in clonal hematopoiesis, MDS and AML, this paper has important implications for understanding how these mutations contribute to pathogenesis. The authors used a complimentary combination of isoform-specific knock-down and over-expression to study the role of individual transcript isoforms in HSC function and mechanistically the effect on DNA methylation and gene expression. The major findings are that DNMT3A2 expression forces HSC differentiation and proliferation, whereas DNMT3A4 inhibits erythropoiesis. Overall this is an interesting study with important implications.

2. Major Points:

- The authors show levels of over-expression by qPCR, but it is important to quantify the level of over-expression from their lentiviruses by Western blot. If the proteins are expressed at supraphysiological levels then the observed effects may be from non-specific effects.
- In Figure 3 - the authors use CD34 expression by FACS as a surrogate marker for number of HSCs in culture over a 7-day period. This immunophenotype is insufficient for HSC identification. The authors should repeat to include more stringent phenotypic definition including CD38⁻ CD45RA⁻ CD90⁺ CD49f⁺.

3. Minor Points:

- Can the authors speculate as to why knock-down of DNMT3A2 induced DNA hypermethylation?

Reviewer #1:

1. *In the study the authors investigate the role of different isoforms of DNMT3a on DNA methylation patterns and gene expression signatures in human stem progenitor cells (CD34+) isolated from the umbilical cord blood. To investigate the function of individual DNMT3a variants the study employs shRNA-mediated knock down (KD) of isoform-specific exons and a lentiviral DNMT3a isoform over-expression (OE) strategy. Overall the study is well designed and addresses the important question if different DNMT3a isoforms show target specific functions, and I am very positive about looking at another version that addresses the following points.*

We thank the reviewer for this encouraging feedback on our work.

2. *The study would strongly benefit if the authors would show the protein expression levels of the various isoforms in HSPC steady state. From the presented data it is unclear how the different isoforms are expressed in the cells of interest (this is also not visible from the RT-qPCRs). The information on variant abundances should be also included for the KD and OE strategies, rather than showing relative % of expression levels in Fig. 1A and B. If western blot analysis are not feasible due to low cell numbers, the expression levels of the individual transcript should be shown relative to a house keeping gene for all experiments.*

This important issue has been raised by both reviewers. We have now isolated new HSPCs, induced knockdown and overexpression of the various transcripts, and expanded them for Western blot analysis as discussed below. Furthermore, as suggested, the expression levels of the individual transcripts in RT-qPCR are now shown relative to GAPDH (new Figure 1B and C). The previous representation with relative % of expression levels has now been moved to the supplement (new Figure S1A and B).

3. *However, Western blots that indicate the levels of all protein isoforms (using general or isoform-specific antibodies) would be very helpful to assess if in the KD/OE experiments the other protein forms would change in their abundance due to changes in isoform stoichiometries.*

Western blot analysis was hampered by the very limited material (despite expansion for 13 days) and by the available antibodies. We utilized two antibodies: N-terminal (rabbit anti-DNMT3A, 2160, Cell Signaling Technology) to detect Tr. 1+3 and Tr. 4; and C-terminal antibody (rabbit anti-DNMT3A, ap1034a, Abgent) to detect Tr. 2. Due to the limited material we often had to load the entire samples on the gels. Either way, the knockdown experiments did not provide enough material for reliable detection. For overexpression, only the N-terminal antibody provided clear results (while the C-terminal antibody gave multiple bands that appeared to be more likely unspecific). These results are now demonstrated in the new Supplemental Figure S1C. The results substantiate overexpression of Tr. 1+3 and Tr. 4, also on protein level, albeit the protein amount differs between the lanes. Overall, it is difficult to judge the changes in isoform stoichiometries due to the aforementioned limitations of our Western blot analysis.

4. *Interestingly, the study identifies several differentially methylated CpGs upon KD of DNMT3A1 and 3A2. While there is minimal overlap for the detected hyper- and hypo-methylated CpGs in the distinct KD samples (Fig. 1G) the authors could also cross-compare the overlap between hypermethylated CpGs after DNMT3a1 KD compared to hypomethylated CpGs in DNMT3a2 KD and vice versa. I am curious to see if the CpGs that reduce methylation in the DNMT3A1 KD overlap with the CpGs that gain in the DNMT3A2 KD - this could potentially hint to a gain of function of the longer isoform.*

As suggested, we have performed the analysis and included it in the Supplemental Figure S2. There is very little or no overlap between these categories, indicating that the knockdown of one splice variant is not leading to a gain of function of the other.

5. *In figure 1K & L, it is hard to see how significant these differences are, especially for H3K9me3 and H3K27me3 where the effect size seems very low according to the axis labels. I agree that the p values indicate statistical significance, but given the high statistical power used in the test, these are meaningless. I suggest removing the p values altogether and instead showing an additional density for foreground signals for the respective modification (i.e. over all detected peaks). Alternatively, a ranked heat map representation, rather than average densities could be shown for the regions around the respective CpGs.*

We agree that the previous representation of chromatin enrichment was confusing and that the presentation of p-values was not informative. In the revised manuscript we present enrichment of individual chromatin marks for the signatures as box plots (new Figure 1K) with corresponding estimates for statistical significance in the new Supplemental Table S2. Additionally, we have moved the old Figure 1K and L to the supplement (Figure S3) as this presentation better depicts that enrichment peaks around the differentially methylated CpGs.

6. *In the end of the publications the authors investigate if variant-specific signatures of DNMT3a can be also found in AML samples. The authors claim that they find only significant correlations with DNMT3A2 isoform, still it is very important to include the results for DNMT3A1 throughout entire Figure 4 in order to make this claim.*

Following the reviewers advice the results for DNMT3A1 signatures in AML are included in the new Supplemental Figure S6.

7. *Additionally, for the results where the study investigate the clinical relevance of DNMT3A2 abundance, it would be interesting to see if DNMT3A1 shows the same trend and how the overall DNMT3a abundance changes in FAB classifications and in patients with poor and intermediate cytogenetic risk compared to favorable risk scores.*

As suggested, we have included clinical correlations of DNMT3A1, DNMT3A2 and overall DNMT3A for the FAB classification and cytogenetic risk groups in the new Supplemental Figure S7.

Reviewer #2:

1. *Short summary of the paper: Bozic et al. present a study investigating the role of distinct isoforms of DNMT3A in hematopoietic stem cell differentiation. Given that DNMT3A mutations are recurrently found in clonal hematopoiesis, MDS and AML, this paper has important implications for understanding how these mutations contribute to pathogenesis. The authors used a complimentary combination of isoform-specific knock-down and over-expression to study the role of individual transcript isoforms in HSC function and mechanistically the effect on DNA methylation and gene expression. The major findings are that DNMT3A2 expression forces HSC differentiation and proliferation, whereas DNMT3A4 inhibits erythropoiesis. Overall this is an interesting study with important implications.*

We thank reviewer 2 for appreciating the importance of our research.

2. *The authors show levels of over-expression by qPCR, but it is important to quantify the level of over-expression from their lentiviruses by Western blot. If the proteins are expressed at supraphysiological levels then the observed effects may be from non-specific effects.*

As indicated above, we have worked hard for Western blot analysis with the limited available material. Our results indicate that Tr. 1+3 and Tr. 4 are indeed overexpressed on protein level, while the expression levels do not seem to be extremely supraphysiological. We cannot exclude non-specific effects, and this is mentioned in the manuscript, but the finding that knockdown and overexpression resulted in complementary effects on DNAm level clearly demonstrates transcript-specific differences (Fig 1 D,E vs. H,I).

3. *In Figure 3 - the authors use CD34 expression by FACS as a surrogate marker for number of HSCs in culture over a 7-day period. This immunophenotype is insufficient for HSC identification. The authors should repeat to include more stringent phenotypic definition including CD38- CD45RA- CD90+ CD49f+.*

Apparently, there was a misunderstanding. While we agree that analysis in hematopoietic stem cells (HSCs) would be very interesting, this would hardly be feasible in our experimental setting. In this manuscript we focused on hematopoietic stem and progenitor cells (HSPCs). We did analyze various hematopoietic markers (including CD38 and CD45; Supplemental Figure S5E). However, the well-established CD38- CD45RA- CD90+ CD49f+ immunophenotype for primary HSCs might also be affected by *in vitro* culture expansion (e.g. CD38 expression decays in fast proliferating cells; PMID: 19432817).

4. *Can the authors speculate as to why knock-down of DNMT3A2 induced DNA hypermethylation?*

In fact, the observed hypermethylation upon knockdown of DNMT3A2 is counterintuitive. As suggested, we have further speculated on the potential reasons:

Line 117-123: “The latter is counterintuitive, but in line with preferential hypermethylation in reduced representation bisulfite sequencing (RRBS) data of Dnmt3a-null HSCs (Challen et al, 2011). It is conceivable that other DNMT3A isoforms or even DNMT3B compensate the downregulation of DNMT3A transcript 2 (Challen et al, 2014). On RNA level knockdown of transcript 2 resulted only in a very moderate upregulation of transcript 1+3 and transcript 4 (Figure S1A), but the general impact on the stoichiometry of different methyltransferases or functional mechanisms remains to be elucidated.”

We thank both reviewers for helpful comments that have helped to further improve our manuscript.

November 29, 2018

RE: Life Science Alliance Manuscript #LSA-2018-00153-TR

Prof. Wolfgang Wagner
RWTH Aachen University Medical School
Stem Cell Biology and Cellular Engineering
Helmholtz-Institute for Biomedical Engineering
Pauwelsstrasse 20
Aachen 52074
Germany

Dear Dr. Wagner,

Thank you for submitting your revised manuscript entitled "Variants of DNMT3A cause transcript-specific DNA methylation patterns and affect hematopoiesis". As you will see, the reviewers appreciate the way you've revised your work, and we would be happy to publish your paper in Life Science Alliance pending final revision to follow the suggestion reviewer #1 makes for Fig. 1K.

A. FINAL FILES:

-- High-resolution figure, supplementary figure and video files uploaded as individual files: See our detailed guidelines for preparing your production-ready images, <http://life-science-alliance.org/authorguide>

B. MANUSCRIPT ORGANIZATION AND FORMATTING:

Full guidelines are available on our Instructions for Authors page, <http://life-science-alliance.org/authorguide>

Sincerely,

Andrea Leibfried, PhD
Executive Editor
Life Science Alliance
Meyershofstr. 1
69117 Heidelberg, Germany
t +49 6221 8891 502
e a.leibfried@life-science-alliance.org
www.life-science-alliance.org

Reviewer #1 (Comments to the Authors (Required)):

The authors have addressed all our concerns.

One minor comment remains related to Fig 1K: In order to circumvent potential biases introduced by different GC content or sequencing coverage between the four region sets, the authors should represent the data as fold-enrichment over input rather than reads. Alternatively, they could introduce another panel that shows the read counts for a ChIP-input sample over the same four sets.

Reviewer #2 (Comments to the Authors (Required)):

The authors have sufficiently addressed all previous criticisms

Aachen, December 4th, 2018

Dear Dr. Leibfried,

Thank you for your kind letter concerning our manuscript “Variants of *DNMT3A* cause transcript-specific DNA methylation patterns and affect hematopoiesis” (Ms. No. #LSA-2018-00153-T).

Following the suggestion of reviewer #1 we have now performed additional normalization of the ChIP experiment in Figure 1K and represented it as fold enrichment over the input. The data still indicate enrichment of hypomethylated CpGs upon KD of transcripts 1+3 and transcript 2 at H3K4me1, while hypermethylated CpGs upon KD of transcript 2 are enriched at repressive histone marks. The p-values were adjusted in Supplemental Table S2 and related changes in the manuscript are highlighted in red.

We are looking forward to your decision.

With best regards,

Wolfgang Wagner

December 5, 2018

RE: Life Science Alliance Manuscript #LSA-2018-00153-TRR

Prof. Wolfgang Wagner
RWTH Aachen University Medical School
Stem Cell Biology and Cellular Engineering
Helmholtz-Institute for Biomedical Engineering
Pauwelsstrasse 20
Aachen 52074
Germany

Dear Dr. Wagner,

Thank you for submitting your Research Article entitled "Variants of DNMT3A cause transcript-specific DNA methylation patterns and affect hematopoiesis". I appreciate the introduced change and it is a pleasure to let you know that your manuscript is now accepted for publication in Life Science Alliance. Congratulations on this interesting work.

*****IMPORTANT:** If you will be unreachable at any time, please provide us with the email address of an alternate author. Failure to respond to routine queries may lead to unavoidable delays in publication.*******

DISTRIBUTION OF MATERIALS:

Again, congratulations on a very nice paper. I hope you found the review process to be constructive and are pleased with how the manuscript was handled editorially. We look forward to future exciting

submissions from your lab.

Sincerely,
